

# Biostimulant red seaweed (*Gracilaria tenuistipitata* var. liui) extracts spray improves yield and drought tolerance in soybean

Md. Abdul Mannan[1], Amir Yasmin[1], Umakanta Sarker[2], Nasimul Bari[1], Dipanjoli Baral Dola[1], Hirokazu Higuchi[3], Sezai Ercisli[4], Daoud Ali[5] and Saud Alarifi[5]

[1] Department of Agronomy, Bangabandhu Sheikh Mujibur Rahman Agricultural University, Gazipur, Bangladesh
[2] Genetics and Plant breeding, Bangabandhu Sheikh Mujibur Rahman Agricultural University, Gazipur, Bangladesh
[3] Graduate School of Agriculture, Kyoto University, Kyoto, Japan
[4] Department of Horticulture, Ataturk University, Erzurum, Turkey
[5] Department of Zoology, College of Science, King Saud University, Riyadh, Saudi Arabia

Corresponding author
Umakanta Sarker,
umakanta@bsmrau.edu.bd

## ABSTRACT

Drought has a deleterious impact on the growth, physiology, and yield of various plants, including soybean. Seaweed extracts are rich in various bioactive compounds, including antioxidants, and can be used as biostimulants for improving yield and alleviating the adverse effect of drought stress. The purpose of this study was to evaluate the effect of soybean growth and yield with different concentrations (0.0%, 5.0%, and 10.0% v/v) of water extracts of the red seaweed *Gracilaria tenuistipitata* var. *liui* under well-watered (80% of field capacity (FC) and drought (40% of FC)) conditions. Drought stress decreased soybean grain yield by 45.58% compared to well-watered circumstances but increased the water saturation deficit by 37.87%. It also decreased leaf water, chlorophyll content, plant height, and the fresh weight of the leaf, stem, and petiole. Drought stress decreased soybean grain yield by 45.58% compared to well-watered circumstances but increased the water saturation deficit by 37.87%. It also decreased leaf water, chlorophyll content, plant height, and the fresh weight of the leaf, stem, and petiole. Under both drought and well-watered situations, foliar application of seaweed extracts dramatically improved soybean growth and production. Under drought and well-watered situations, 10.0% seaweed extract increased grain yield by 54.87% and 23.97%, respectively in comparison to untreated plants. The results of this study suggest that red seaweed extracts from *Gracilaria tenuistipitata* var. *liui* may be used as a biostimulant to improve soybean yield and drought tolerance in the presence of insufficient water. However, the actual mechanisms behind these improvements need to be further investigated in field conditions.

## INTRODUCTION

Due to its high protein content (40–42%), oil content (18–22%), and nitrogen-fixing potential (17–127 kg nitrogen ha$^{-1}$ year$^{-1}$), soybean (*Glycine max* L.) is a prominent legume and one of the most significant oilseed crops in the world (*Anderson, Johnstone & Cook-Newell, 1995*). Dry seeds of soybean contain calcium (Ca), iron (Fe), magnesium (Mg), phosphorus (P), potassium (K), and folic acid, as well as several vitamins, including different B vitamins and minerals, such as molybdenum (Mo), copper (Cu), manganese (Mn) (*Banaszkiewicz, 2011*). The edible vegetable oil and high-protein feed supplements for livestock, poultry, and aquaculture are primarily supplied by this crop. Other fractions and derivatives of soybean seeds have been used to make a wide range of industrial, food, medicinal, and agricultural products, all of which have a significant economic influence (*Smith & Huyser, 1987*).

From a total land area of 130.90 million hectares, 365.79 million tons of soybeans are produced annually in the world (*FAOSTAT, 2017*). Bangladesh produces 96,921 tons of food annually from a cultivable area of 62,870 hectares; this amounts to 1.54 tons per hectare, which is significantly less than the 2.79 tons per hectare average for the world. Brazil is the largest producer of soybeans in the world followed by the United States and Argentina. Soybean is called the "golden bean" and "miracle crop" of the 21st century (*Tambe, Pedhekar & Harshali, 2021*) and the demand for soybean is increasing day by day in Bangladesh. This is due to public awareness of its high nutritional value and its use as an ingredient in poultry, livestock, and fish feed (*Haque et al., 2020*; *Dola et al., 2022*). However, in the drought-prone north-western regions and lands of coastal areas of Bangladesh, soybean is one of the most competitive crops for farmers and finds its place in marginal lands (*Dola et al., 2022*). Recently, as a result of the global climate, droughts have become worse and more common.

Plants are rich in biochemicals, phenolics, and antioxidants (*Sarker & Ercisli, 2022*; *Binici, Sat & Aoudeh, 2021*; *Kurubas, Sabotic & Erkan, 2021*; *Sokolova et al., 2021*) flavonoids (*Sarker & Oba, 2020a*, *2020b*, *2020c*), pigments (*Sarker & Oba, 2021*; *Sarker et al., 2022a*), minerals (*Chakrabarty et al., 2018*; *Sarker, Azam & Talukder, 2022*), protein (*Sarker et al., 2014*), dietary fiber, carbohydrates, and vitamins (*Sarker, Hossain & Oba, 2020*; *Sarker, Oba & Daramy, 2020*; *Sarker et al., 2020*) for human nutrition. The majority of them are secondary metabolites that serve as nonenzymatic antioxidants and secondary nutrient sources, protecting plants from the damaging effects of any abiotic stress.

Stressors such as drought and salinity affect crop productivity due to the generation of reactive oxygen species (ROS) (*Sarker & Oba, 2018a*). These ROS cause oxidative damage and osmotic stress in plants (*Sarker & Oba, 2018b*), membrane, DNA, and protein damage, and nutrient imbalances (*Sarker & Oba, 2018c*, *2018d*), as well as a decrease in changes in color pigments during photosynthetic activity (*Sarker et al., 2018a*; *Hossain et al., 2022*). Soybeans under water-deficient conditions can experience several physiological and biochemical changes (*Dola et al., 2022*), including reduced leaf water status (*Mannan et al.,*

*2021*), $CO_2$ assimilation, gas exchange rate, and chlorophyll synthesis (*Ferdous et al., 2018*), which ultimately has a destructive effect on plant growth, production, and quality.

Plant growth and yield has been improved by biostimulants as a result of biological approaches under stressful conditions (*Rouphael & Colla, 2020*). Foliar application of seaweed extracts is one of eight major classes of biostimulants that have shown significant improvement in stress tolerance in a variety of crops, including cereals, flowers, grasses, and various vegetables. Seaweed extracts contain a variety of nutrients like P, Ca, Mg, and Fe, secondary metabolites, and other biochemicals that have beneficial effects on plants, including yield enhancement (*Rouphael & Colla, 2020*). Seaweed extract attenuating the adverse effects of drought, cold, and salinity was found to be mediated by the accumulation of non-structured carbohydrates that enhance energy storage, metabolism, and water regulation, as well as proline accumulation (*Dalal et al., 2019*; *Xu & Leskovar, 2015*).

Among many countries, Bangladesh has started growing *Gracilaria tenuistipitata* var. *liui* to produce jelly. Previous studies have shown that the use of seaweed extracts stimulates plant growth and mitigates abiotic stress (*Di Mola et al., 2019*; *Shukla et al., 2018*; *Mattner, Milinkovic & Arioli, 2018*; *Rouphael et al., 2018*), to date no studies have been conducted on the effects of red algae *Gracilaria tenuistipitata* var. *liui* on growth, yield, and drought tolerance of plants. We hypothesize that the aqueous extract of *Gracilaria tenuistipitata* var. *liui*, can improve the drought tolerance of soybean. Therefore, the aim of this study was to assess the effects of foliar application of *Gracilaria tenuistipitata* var. *liui* extracts as a biostimulant on soybean growth, physiology, and yield under conditions of water stress.

## MATERIALS AND METHODS

### Experimental site

The pot experiment was conducted in a controlled environment in a polythene indoor at the Department of Agronomy, Bangabandhu Sheikh Mujibur Rahman Agricultural University (BSMRAU), Bangladesh (24°5′23″N and 90°15′36″E) during November 2019 to March 2020 in *Rabi* season. The mean temperatures were 28.5 ± 1.6 °C and 13.6 ± 1.3 °C, at daytime and nighttime, respectively and relative humidity was 60–70% during experimentation (*Bangabandhu Sheikh Mujibur Rahman Agricultural University (BSMRAU), 2020*). The plants were grown in plastic pots (0.30 m deep and 0.25 m in diameter), each containing 11 kg of soil. The pH and moisture content in the experimental soil at field capacity were 6.71% and 28%, respectively and the texture was sandy loam (53.12% sand, 33.12% alluvium, and 13.76% clay). In dry soil, electrical conductivity (EC), cation exchange capacity (CEC), total N, K, available P, and organic carbon were 0.03 dS/m, 12.85 cmol/kg, 0.06%, 0.76 cmol/kg, 0.07 mg/100 g and 0.59%, respectively.

### Plant material and seaweed extracts

The soybean variety BU soybean-1 was grown in the pots. From the coastal area of the Moheshkhali Channel of the Bay of Bengal (21°30′0″N and 92°5′0″E) in Bangladesh seaweed, *Gracilaria tenuistipitata* var. *liui* (red algae) (family *Gracilariaceae*) was collected. The collected fresh seaweeds were kept in the laboratory at room temperature (15–20 °C)

**Table 1 Chemical constituents of *Gracilaria tenuistipitata* var. *liui* seaweed extracts.**

| Constituents | Concentration |
| --- | --- |
| Crude protein (%) | 24.46 ± 0.10[*] |
| Crude fiber (%) | 5.05 ± 0.13 |
| Crude lipid (%) | 0.15 ± 0.02 |
| Carbohydrates (%) | 48.45 ± 0.45 |
| Ash (%) | 10.22 ± 0.14 |
| Moisture (%) | 11.68 ± 0.09 |
| Phosphorus (mg/100 g dry weight) | 580.65 ± 6.36 |
| Calcium (mg/100 g dry weight) | 130.64 ± 1.07 |
| Magnesium (mg/100 g dry weight) | 3.40 ± 0.21 |
| Iron (mg/100 g dry weight) | 76.58 ± 0.26 |
| Copper (mg/100 g dry weight) | 3.89 ± 0.40 |
| Pb (mg/kg dry weight) | 0.041 ± 0.02 |
| β-carotene (mg/100 g) | 10.21 ± 0.58 |
| Vitamin C (mg/100 g) | 2.72 ± 0.41 |
| Total energy (kcal/100 g) | 300.30 ± 0.89 |

Note:
[*] Mean ± standard error (SE).

and immediately washed with seawater and tap water to remove the unwanted impurities. Then the seaweed dried in the sun. The dried seaweed was ground by a grinder with stainless steel blades at ambient temperature (15–20 °C) and immediately utilized for extraction of liquid fertilizer. Following the method described by *Eswaran et al. (2005)*, seaweed powder was utilized for the extraction of liquid fertilizer. A total of 50 and 100 g of seaweed powder were added to 1 L of distilled water to prepare 5.0% and 10.0% seaweed extracts solution, respectively in separate two beakers. To mix the solute properly both of the solutions were heated at 60 °C temperatures for 45 min on a magnetic stirrer with a hot plate and at room temperature (15–20 °C) the solutions were stored in different two plastic bottles for 1 h until application. During the application of the solution into the plant, the required amount of solution was inserted into the hand sprayer. Following different methods, the compositions of the seaweed were determined and are presented in Table 1.

## Determination of crude protein and lipid

The crude protein content was determined by the Micro-Kjeldahl method (*Sarker et al., 2022b*; *Guebel, Nudel & Giulietti, 1991*), and the crude lipid content of seaweed was determined using the method by *Mehlenbacher (1960)*.

## Determination of crude fiber

The crude fiber content was determined using the AOAC method (*Association of Official Analytical Chemists (AOAC), 2000*). A total of 200 mL of 0.255 N sulfuric acid was added to the moisture and fat-free 5.0 g sample and boiled for 30 min before adding 200 mL of 0.313 N (1.25%) NaOH solution and boiling for another 30 min. The filtrate was weighed after drying at room temperature. The sample was then placed in a muffle furnace at

650 °C for 2–3 h before being cooled and weighed again. The weight difference represents the amount of crude fiber.

## Determination of moisture

To measure the moisture content 1 g of seaweed powder was placed onto the tray of the automatic moisture meter (Model PB-1D2, 544205; Kett Electric laboratory, Tokyo, Japan) for 10–15 min.

## Determination of ash

Ash content was determined by following AOAC method (*Association of Official Analytical Chemists (AOAC), 1990*). About 8 g of finely ground dried sample was weighed into a porcelain crucible and incinerated at 550 °C for 6 h in an ashing muffle furnace until ash was obtained. The ash was cooled in desiccators and reweighed. The % ash content in the seaweed sample was calculated using the formula: Ash (%) = (Weight of ash/weight of a sample taken) × 100.

## Determination of carbohydrate

The percentage of total carbohydrate content was calculated using the formula

Percentage of total carbohydrate content = 100 − (% moisture + % crude fiber + % crude protein + % crude lipid + % ash) described by *Sarkiyayi & Agar (2010)*.

## Determination of available energy

Available energy = [(9 × fat) + (4 × carbohydrates) + (4 × protein)] was calculated following the formula described by *Eneche (1991)* and *Tarafder et al. (2023)*.

## Determination of minerals, heavy metals, β-carotene, and vitamin C

Atomic absorption spectrophotometer (Model- AA. 610s; Shimadzu, Kyoto, Japan) was used to determine the mineral contents (Ca, Mg, Fe, Cu, and P) and heavy metal (Pb) following Hitachi, Ltd. (*Hitachi LTD, 1986*). The amount of β-carotene in the sample was quantified using visible spectroscopy following the methods of *Hassan et al. (2022a)* and *Sarker et al. (2023)*. Vitamin C was quantified using the method described by *Hassan et al. (2022b)*.

## Treatments and cultural practices

There are two factors in the experiment. Factor 1: i. well-watered (80% of field capacity (FC)) (Control), ii. Drought (40% of FC), Factor 2: three doses of seaweed extract (0.0%, 5.0%, and 10.0% v/v). In each pot, 10 soybean seeds were sown on 23 December 2020 and to ensure uniform germination the pots were irrigated thoroughly. In each pot, six healthy seedlings were kept when the seedlings were fully established. Urea, superphosphate, and potassium chloride were applied in the pot at 0.27, 0.28, and 0.20 g ((equivalent to 60, 75, 120 kg/ha) (*Bangladesh Agricultural Research Council, 2018*)). Regular irrigation was provided throughout the growing season after fourteen days of sowing of seeds to maintain 80% FC in nine pots and 40% FC in the other nine pots. Pots were inspected at regular intervals to determine soil moisture content using a portable POGO Soil Sensor II digital

moisture meter (Stevens, Hoboken, NY, USA). Seven days after imposition of drought (2nd trifoliate stage), all the pots were sprayed with different doses of seaweed extracts using a hand pressure sprayer (Seesa-Pump & Spray, GA-013, the spraying tip- Mist Nozzle Set with T Joint for Foggy Water 8 mm Pipe). The plants were sprayed four times at two-week intervals throughout the growing season. A total of 50–60 mL of algae extract was sprayed on three plants in pots. Experiments were performed using a randomized complete block (RCBD) design containing three replicates.

## Growth and agronomic measurement

Growth-related parameters viz. plant height, fresh weight of leaf, petiole, and stem were measured at the flowering stage (15 days after 1st spraying). At the physiological maturity stage dated 01 April 2021, plants were harvested and data on the number of pods/plant, the number of seeds/pod, 100-seed weight (g), and seed yield/plant (g) were recorded.

## Chlorophylls in leaf

Fully developed leaves from the top were sampled replication-wise at the flowering stage. *Sarker et al. (2022c, 2022d, 2022e)* methods were followed to estimate the chlorophyll content. In a test tube, a 20 mg fresh leaf sample was extracted with 20 mL of 80% acetone and stored in the dark for 72 h. A double-beam spectrometer (Thermo Fisher Scientific, Waltham, MA, USA) was used to take the readings at 663 and 645 nm. The results were expressed as mg/g fresh weight.

## Leaf water status

Relative water content (RWC) and water saturation deficit (WSD) of soybean leaves were calculated during flowering (15 days after first spraying). The fully developed uppermost fresh leaf was weighed immediately. Then, distilled water was utilized to soak the leaves for 24 h in the dark at room temperature (15–20 °C). After that, the excess water was wiped out with a article towel and the turgid weight of the leaves was taken. To measure their dry weight the leaves were dried later in an oven for 48 h at 72 °C. RWC and WSD were calculated using the formula of *Schonfeld et al. (1988)*: relative water content (RWC) and water saturation deficit (WSD) of soybean leaves were calculated during flowering (15 days after first spraying). The fully developed uppermost fresh leaf was weighed immediately. Then, distilled water was utilized to soak the leaves for 24 h in the dark at room temperature (15–20 °C). After that, the excess water was wiped out with a article towel and the turgid weight of the leaves was taken. To measure their dry weight the leaves were dried later in an oven for 48 h at 72 °C. RWC and WSD were calculated using the formula of *Schonfeld et al. (1988)*:

$$RWC\ (\%) = \frac{FW - DW}{TW - DW} \times 100, \quad WSD\ (\%) = \frac{TW - FW}{TW - DW} \times 100$$

## Statistical analysis

To obtain a replication mean, we averaged each treatment from all the sample data of a trait (*Rashad & Sarker, 2020*). We biometrically and statistically analyzed the mean data of

**Table 2 Effect of seaweed extracts on plant height and leaf fresh weight of soybean at flowering stage under well-watered and drought conditions.** Lowercase letters indicate that mean values are significantly different from one another.

| Seaweed extracts | Plant height (cm) | | Leaf fresh weight/plant (g) | |
|---|---|---|---|---|
| | Well-watered | Drought | Well-watered | Drought |
| 0.0% | 28.70c ± 1.01[*] | 23.50e ± 1.20 | 11.10b ± 0.10 | 6.41d ± 0.24 |
| 5.0% | 31.50b ± 0.92 | 26.96d ± 0.50 | 11.28ab ± 0.08 | 7.54c ± 0.27 |
| 10.0% | 33.80a ± 0.80 | 31.66b ± 1.01 | 11.84a ± 0.83 | 7.97c ± 0.21 |
| Level of significance | ** | | ** | |
| CV (%) | 3.2 | | 4.1 | |

**Notes:**
[*] Mean values ± SE.
[**] for $p < 0.01$.
CV, coefficient of variation.

**Table 3 Effect of seaweed extracts on petiole fresh weight, stem fresh and total fresh weight of soybean at flowering stage under well-watered and drought conditions.** Lowercase letters indicate that mean values are significantly different from one another.

| Seaweed extracts | Petiole fresh weight/plant (g) | | Stem fresh weight/plant (g) | | Total fresh weight/plant (g) | |
|---|---|---|---|---|---|---|
| | Well-watered | Drought | Well-watered | Drought | Well-watered | Drought |
| 0.0% | 2.55a ± 0.14[*] | 1.16c ± 0.07 | 5.30b ± 0.10 | 2.42d ± 0.45 | 18.96b ± 0.15 | 9.99e ± 0.34 |
| 5.0% | 2.65a ± 0.36 | 1.62b ± 0.11 | 5.57ab ± 0.21 | 3.63c ± 0.27 | 19.51b ± 0.51 | 12.79d ± 0.32 |
| 10.0% | 2.85a ± 0.11 | 1.72b ± 0.11 | 5.80a ± 0.10 | 3.92c ± 0.07 | 20.50a ± 0.85 | 13.62c ± 0.18 |
| Level of significance | ** | | ** | | ** | |
| CV (%) | 8.5 | | 5.4 | | 2.9 | |

**Notes:**
[*] Mean values ± SE.
[**] for $p < 0.01$.
CV, coefficient of variation.

various traits (*Hasan-Ud-Daula & Sarker, 2020*). Statistix 8 software was used to analyze the data for obtaining an analysis of variance (ANOVA) (*Hasan et al., 2020*, *2022*). The experiment was carried out with two factors. We have analyzed the data for the main effects and interaction effects of two factors following the factorial randomized completely block design. All treatment means were compared at the 1% significance level using the least significant difference (LSD) test.

# RESULTS

## Growth parameters

Water stress reduced plant height, leaves, petioles, stems, and total fresh weight by 18.12%, 42.25%, 54.50%, 54.34%, and 47.31%, respectively, compared with control plants (well-watered) (Tables 2 and 3). When plants were sprayed with seaweed extract, they grew taller in drought conditions by 34.72% compared to drought plants. However, when plants were sprayed with the same concentration of seaweed extract in well-watered conditions, the increment was 17.77% in comparison to untreated control plants. Leaf fresh weight, petiole fresh weight; stem fresh weight, and total fresh weight increased by 24.34%, 48.28%, 61.98%, and 36.34%, respectively, under water stress conditions, when plants were sprayed

**Table 4 Effect of seaweed extracts on chlorophyll *a*, chlorophyll *b*, and total chlorophyll content of soybean leaf at flowering stage under well-watered and drought conditions.** Lowercase letters indicate that mean values are significantly different from one another.

| Seaweed extracts | Chlorophyll *a* (mg/g FW) | | Chlorophyll *b* (mg/g FW) | | Total chlorophyll (mg/g FW) | |
|---|---|---|---|---|---|---|
| | Well-watered | Drought | Well-watered | Drought | Well-watered | Drought |
| 0.0% | 1.42 ± 0.03* | 1.21 ± 0.02 | 1.08 ± 0.03 | 0.9 ± 0.10 | 2.50c ± 0.02 | 2.11d ± 0.08 |
| 5.0% | 1.51 ± 0.03 | 1.47 ± 0.03 | 1.17 ± 0.03 | 1.2 ± 0.04 | 2.68b ± 0.05 | 2.68b ± 0.01 |
| 10.0% | 1.60 ± 0.05 | 1.57 ± 0.03 | 1.30 ± 0.04 | 1.26 ± 0.03 | 2.90a ± 0.09 | 2.83a ± 0.06 |
| Level of significance | NS | | NS | | ** | |
| CV (%) | 2.2 | | 4.2 | | 2.2 | |

Notes:
* Mean values ± SE.
** for $p < 0.01$.
FW, fresh weight; NS, non-significant; CV, coefficient of variation.

with a 10.0% concentration of seaweed extract compared with untreated plants. Elsewhere, under well water conditions, the increments of these parameters were 6.66%, 11.76%, 9.43%, and 8.12%, respectively, when plants were treated with the same concentration of seaweed extracts.

## Physiological traits

### Chlorophyll content

The lowering trend of chlorophyll content in plant leaves was also caused by the water stress condition in soybean plants. This stressed condition resulted in a decrease of 14.79% chlorophyll *a*, 16.67% chlorophyll *b*, and a total of 15.60% chlorophyll content in soybean leaves (Table 4). Leaf chlorophyll values increased after spraying with seaweed extracts, suggesting that foliar applications of seaweed extracts alleviated the negative effects of water deficit. In the 5.0% treatment, chlorophyll *a*, chlorophyll *b*, and total chlorophyll increased by 6.34%, 8.33%, and 7.20%, respectively compared to untreated plants under well-water conditions. On the other hand, these increments were 21.49%, 33.33%, and 27.01% compared to untreated plants under drought conditions. The maximum chlorophyll content was obtained through spray with 10.0% concentrated seaweed extract on exogenous leaves. This application of seaweed extracts improved chlorophyll *a* by 29.75%, chlorophyll *b* by 40.0%, and total chlorophyll by 34.12% compared to untreated plants under drought conditions. In addition, exogenous leaf seaweed extracts increased chlorophyll *a* by 12.68%, chlorophyll *b* by 20.37%, and total chlorophyll by 16.00% compared to untreated plants.

### Leaf water status

Water stress significantly reduced the relative water content (RWC) by 16.0% compared with the well-watered condition (Fig. 1). Seaweed extract applied to *Gracilaria tenuistipitata* var. *liui* at different concentrations promoted RWC in soybean leaves grown under control and water deficit conditions. With the application of seaweed extract at 10.0% concentration, RWC increased by 12.36% in drought-stressed plants and 6.32% in well-watered plants compared with untreated plants. In 5.0% treatment, RWC increased
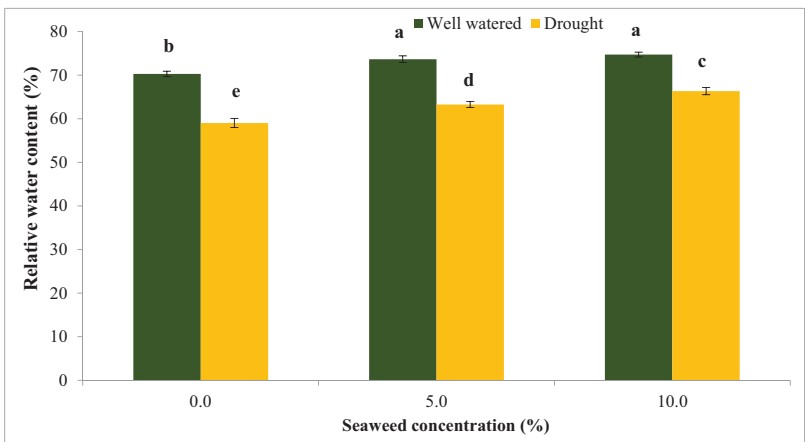

**Figure 1 The influence of seaweed extracts on the leaf water content of soybean under well-watered and drought conditions. Bars indicate (±standard error).**

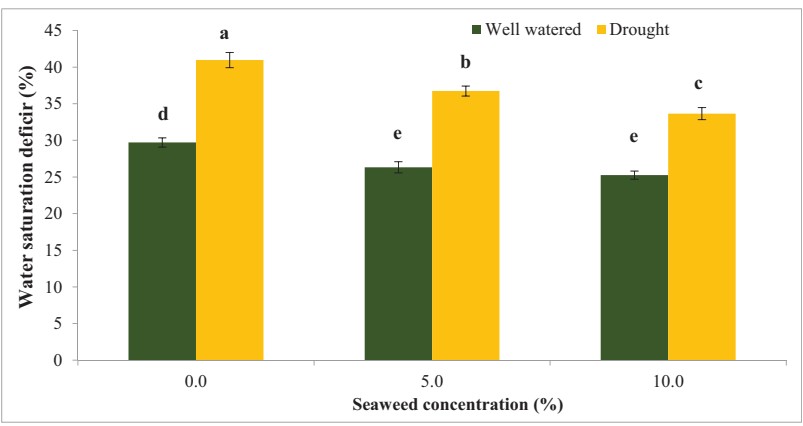

**Figure 2 Effect of seaweed extracts on water saturation deficit (WSD) of soybean leaf at flowering stage under well-watered and drought conditions. Bars indicate (±standard error).**

by 4.80% compared to untreated plants under well-watered conditions and it was 7.15% under drought-stressed plants. In contrast, water deficit significantly increased water saturation deficit (WSD) by 37.87% in drought-stressed plants compared with well-watered plants (Fig. 2). However, the foliar spray of seaweed extract notably progressed the water saturation deficit in plants under well-watered and drought situations. In 5% seaweed extract treatment, the WSD was reduced by 11.41% in well-watered conditions, whereas this reduction was 10.32% under drought conditions. The WSD was reduced by 14.98% in well-watered plants, whereas it was 17.85% in drought-stressed plants at a 10.0% concentration of seaweed extract. Relative water content (RWC) refers to the water status of plant leaves. The outcomes of this test displayed that RWC notably declined because of drought stress. Besides that, drought-pressured vegetation exhibited first-rate development in WSD relative to non-pressured ones. However, the exogenous spray of seaweed extracts ensured superior

**Table 5 Effect of seaweed extracts on number of pods/plant, number of seeds/pod, 100-seed weight and seed yield of soybean under control and drought conditions.** Lowercase letters indicate that mean values are significantly different from one another.

| Seaweed extracts | Number of pods/plant | | Number of seeds/pod | | 100 seed weight (g) | | Seed yield/plant (g) | |
|---|---|---|---|---|---|---|---|---|
| | Well-watered | Drought | Well-watered | Drought | Well-watered | Drought | Well-watered | Drought |
| 0.0% | 52.02b ± 0.37* | 37.44e ± 1.02 | 1.93b ± 0.06 | 1.66c ± 0.07 | 10.30c ± 0.18 | 7.82f ± 0.13 | 10.18c ± 0.16 | 5.54f ± 0.41 |
| 5.0% | 53.51b ± 0.61 | 46.70d ± 1.48 | 2.06a ± 0.12 | 1.77c ± 0.07 | 11.24a ± 0.27 | 8.54e ± 0.24 | 11.89b ± 0.17 | 6.71e ± 0.24 |
| 10.0% | 56.43a ± 0.77 | 48.76c ± 1.04 | 2.13a ± 0.03 | 1.87b ± 0.08 | 11.63b ± 0.12 | 9.59d ± 0.28 | 12.62a ± 0.11 | 8.58d ± 0.24 |
| Level of significance | ** | | ** | | ** | | ** | |
| CV (%) | 1.9 | | 4.0 | | 2.2 | | 2.6 | |

Notes:
* Mean values ± SE.
** for $p < 0.01$.
CV, coefficient of variation.

RWC and decreased WSD below water deficit and well-watered situations relative to untreated vegetation. The maximum enhancement turned assigned for the exogenous spray at a 10.0% concentration of seaweed extracts in comparison to a 5.0% concentration of the identical biostimulant seaweed extracts.

## Yield and its contributing characters

In comparison to the well-watered (control) condition, the number of pods per plant, the number of seeds per pod, and the weight of 100 seeds dramatically decreased under the drought conditions by 28.20%, 13.98%, and 24.08%, respectively. The number of pods per plant, the number of seeds per pod, and the weight of 100 seeds were significantly enhanced by 30.24%, 12.65%, and 22.63%, respectively, when sprayed with seaweed extract at a concentration of 10% in drought-stressed plants as opposed to untreated plants. (Table 5). Similarly, compared to untreated plants grown in well-watered (control) conditions, foliar spraying seaweed extract at the same concentration boosted all three attributes by 8%, 10%, and 12%, respectively. Water shortage had an impact on seed yield per plant, which was reduced by 45.58% in comparison to the control condition (Table 5). The negative effects of water-stressed conditions on seed yield were significantly reduced by the exogenous foliar application of seaweed extract. In comparison to untreated drought-stressed plants, exogenous foliar spray of seaweed extract at a concentration of 10% showed the highest increment in seed yield of 54.87%. However, when the well-watered plant was sprayed with 10.0% seaweed extract, this rise was more than 23.97% compared to the untreated well-watered plants.

## DISCUSSION

The purpose of this research is to determine how soybean drought tolerance is increased by using the red seaweed *Gracilaria tenuistipitata* var. *liui* extracts. When seaweed extracts were sprayed on soybean plants, more vegetative development was observed under water stress conditions compared to controlled settings (Fig. 3). Seedling fresh and dry weights were changed significantly in response to treatment with different concentrations of the seaweed extracts in our study. *Senthuran et al. (2019)* recorded that the extracts of

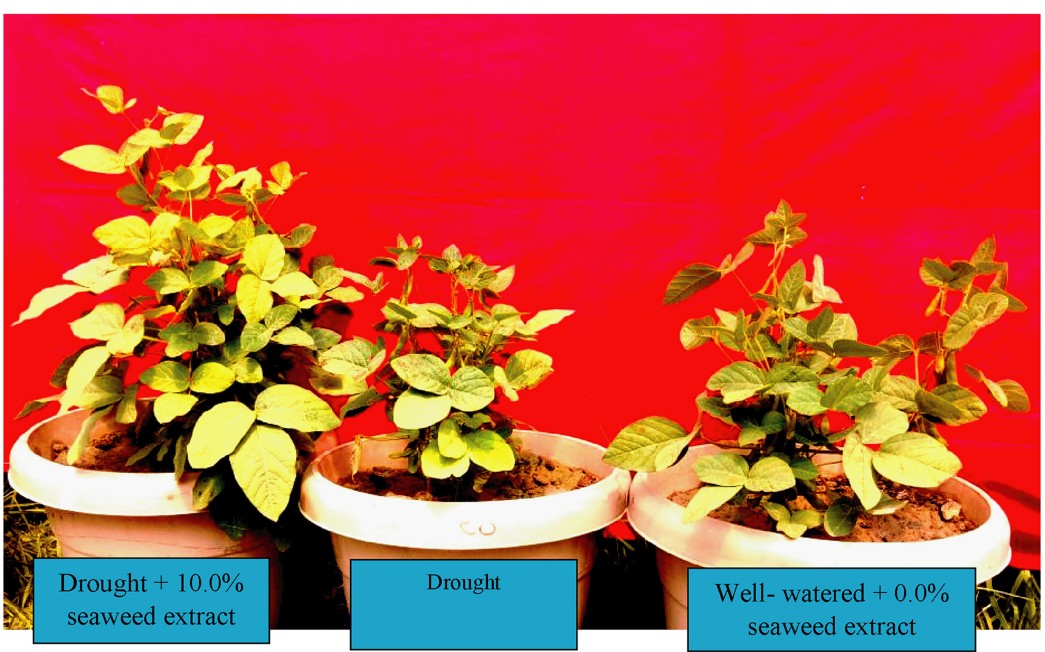

Figure 3 The response of seaweed extracts on growth performance of soybean under well-watered and drought conditions.

*A. nodosum* and *K. alvarezii* also improved water and nutrient uptake, which ultimately led to the promotion of overall vigor and the growth of plants. Seaweed extracts contain amino acids and minerals that actively promote plant growth and development in soybean under drought conditions which are corroborative to the results of *Tarakhovskaya, Maslov & Shishova (2007)*. Seaweed contains macro and micronutrients (*Kalaivanan & Venkatesalu, 2012*), vitamin C and β-carotene may also contribute to the ability of treated plants to promote growth (*Abeed et al., 2021*; *Huda et al., 2023*). Plant growth promotion by applying seaweed extracts has also been reported in strawberries (*Alam et al., 2013*). Our results were consistent with previous studies on *Cajanus cajan* (*Mohan et al., 1994*), *Vigna sinensis* (*Sivasankari et al., 2006*), and *Zea mays* (*Al-Shakankery, Hamouda & Ammar, 2014*). The beneficial effects of the application of seaweed extract may be due to improved root growth and settlement which ultimately help to absorb more nutrients from deeper layers of the soil in a balanced ratio. However, this study demonstrates for the first time that the seaweed extract *G. tenuistipitata* var. *liui* from the Bay of Bengal has greatly improved the growth of an economically important crop, soybeans. Literature has shown that seaweed extracts increased the number of leaves, height, and leaf width in all stages of plant growth until harvesting (*Ali, Ramsubhag & Jayaraman, 2021*). *Naz & Bano (2013)* displayed that the macro and micronutrients in *C. procera* extracts were easily absorbed by the target plants and played an important role in the plant's vital metabolism like glycolysis. These nutrients have been shown to have growth-promoting effects on maize crops.

The growth regulatory substances (seaweed oligosaccharides as carbohydrates) induced the biosynthesis of hormones such as phytohormones abscisic acid, cytokinin, and auxin in

treated plants (*Aremu et al., 2016*) that could promote crop growth. Phytohormones salicylic acid (SA), 1-aminocyclopropane-1-carboxylic acid (ACC), and Zeatin stimulate growth and development, such as root hair proliferation, cell division, water balance regulation, and stomatal conductance under drought stress (*Tsang et al., 2011*). The augmented buildup of SA and cytokinins by different biostimulant Plant growth-promoting rhizobacteria (PGPR) strains under drought stress has been reported in previous studies (*Jochum et al., 2019*). PGPR Biostimulant-mediated drought-tolerance and growth mechanisms are associated with the greater expression of drought-stress-responsive defense genes and the expression of key genes which regulate increased plant biomass (*Lephatsi et al., 2022*). Application of *Kappaphycus alvarezii* seaweed extract in maize under drought up-regulated the expression of genes related to the enhancement of auxin and gibberellic acid signaling, root growth, seed development, transport, nitrogen metabolism, and antioxidant activity like peroxidases and glutathione *S*-transferase compared to its control (*Kumar et al., 2020*). *Zhang & Schmidt (1999)* observed enhanced antioxidant status after foliar application of seaweed extract under water deficit conditions in Kentucky bluegrass. High molecular weight biostimulants (algal polysaccharides) in seaweed extracts could improve crop stress tolerance in tall fescue and creeping bentgrass (*Zhang & Schmidt, 2000*). We estimated that water deficits reduced chlorophyll pigments in soybean leaves under drought conditions. *Youssef et al. (2018)* reported that seaweed extract applications were correlated with increased biosynthesis of chlorophyll (higher SPAD index) due to magnesium constituent, which was necessary for chlorophyll synthesis in plants. Our results showed that seaweed extract spray could significantly increase the total chlorophyll content of soybean leaves, which was also confirmed by other reports (*Anjum et al., 2011*; *Yang et al., 2012*). This might be probably because of the presence of essential, and nonessential amino acids, and different energetic materials in seaweed extract that inhibited the deterioration of chlorophylls (*Blunden, Jenkins & Liu, 1996*) under drought stress. In our research, we determined that the use of seaweed extracts elevated the hydration repute of soybean leaves. *Abeed et al. (2021)* found that different kinds of biostimulants sprayed on plants significantly increased the production of primary metabolites in terms of carbohydrates, and amino acids. As a result, investment in plant biomass may be related to the enhancement of chlorophyll content in the plant, which may increase the photosynthetic rate and stimulate the source-to-sink transport of sugars, increasing carbohydrate content. These results were in agreement with those of *Lashin et al. (2013)*, who demonstrated that the treatment of cowpea (*Vigna unguiculata*) with aqueous extracts of *Malva parviflora* L. and *Artemisia ludia* L. considerably enhanced the yield and growth. According to *Du Jardin, Xu & Geelen (2020)*, the use of plant biostimulants such as seaweed extract increased nutrient use efficiency, tolerance to abiotic stress, and crop quality.

In the current study, we observed that seaweed extract had more iron content (Table 1), which also brought about more potent osmotic alteration, a better relative content of water, and progressed membrane balance with the aid of augmenting the concentrations of osmoregulation in plant cells. Seaweed extract-induced reduction of the detrimental effect of water-stressed conditions had been demonstrated to be mediated by improved root

morphology, a build-up of non-structural carbohydrates that increased energy storage, accelerated metabolism, and water adaptations (*Dalal et al., 2019*). The increase in yield of soybean by the extracts of *Gracilaria tenuistipitata* var. *liui* has not been reported earlier.

In our experiment, the application of seaweed extracts increased the morphological growth, number of pods per plant, number of seeds per pod, and 100-seed weight which raised the overall production of seed yield, which was supported by previous researchers (*Rathore et al., 2009*; *El Modafar et al., 2012*; *Gajc-Wolska, Spiżewski & Grabowska, 2013*). Similarly, *Rama Rao (1991)* reported an improvement in yield and enhancement in the quality of *Zizyphus mauritiana* Lamk through the exogenous spray of seaweed extract. *Dookie et al. (2021)* and *Ali, Ramsubhag & Jayaraman, 2021* reported that early flowering triggered and increased fruit set in tomatoes and peppers by application of seaweed extract. This increase in the number of flowers and fruit sets inevitably leads to an improvement in yield. In addition, flower number, flower/fruit ratio, fruit number, and tomato size were improved by applying seaweed extracts (*Di Stasio et al., 2020*). Seaweed (*Ascophyllum nodosum*) extracts treated soybean plants had higher stomatal conductance relative water content and antioxidant activity under drought stress. In addition, *A. nodosum* treatment led to changes in the expression of stress-responsive genes, such as *GmCYP707A3b*, *GmCYP707A1a*, *GmRD22*, *GmDREB1B*, *GmRD20*, *GmERD1*, *FIB1a*, *GmNFYA3*, *GmPIP1b*, *GmGST*, *GmTp55* and *GmBIP* (*Shukla et al., 2018*). The presence of various levels of phytohormones such as cytokinins and the induction of host hormonal synthesis in the seaweed extracts can be responsible for increasing the yield (*Sakakibara, 2006*; *Farber, Attia & Weiss, 2016*). Iron is one of the important micronutrients in seaweed extract that minimize the adverse effects caused by water shortage conditions in soybean (*Deswal & Pandurangam, 2018*). Therefore, it appears that foliar applications of seaweed extracts are effective in mitigating the adverse effects due to water scarcity.

## CONCLUSION

Drought stress significantly reduced all physiological, morphological, and agronomic parameters of soybean plants compared to well-watered plants. Conversely, foliar applications of seaweed extracts reduced the adverse effects of water scarcity. Foliar application of 10.0% seaweed extracts decreased drought stress more successfully than other treatments. Crop dry matter accumulation, yield components, and grain yield were all positively affected by the application of seaweed extract. Finally, this study indicates that the extract of *G. tenuistipitata* var. *liui* promotes growth, yield, and drought tolerance in soybean. From the current results, it was concluded that foliar spray of seaweed extract can be suggested to help soybean production in drought-prone areas to minimize the adverse effects of drought through better adaptation to water deficit stress. It is a controlled experiment. Therefore, additional research is required to assess the impact of these extracts under field conditions. Further study is required to determine whether the extract of *G. tenuistipitata* var. *liui* affects the expression of molecular characteristics such as peroxidase (POD), superoxide dismutase (SOD), and catalase (CAT) in soybean.

## ACKNOWLEDGEMENTS

We appreciate Dr. S. M. Rafiquzzaman for providing the lab space to prepare seaweed extracts at the Department of Fisheries Biology and Aquatic Environment, BSMRAU.

### Funding

This research was funded by Researchers Supporting Project number (RSP2023R27), King Saud University, Riyadh, Saudi Arabia. The funders had no role in study design, data collection and analysis, decision to publish, or preparation of the manuscript.

### Grant Disclosures

The following grant information was disclosed by the authors:
Researchers Supporting Project, King Saud University, Riyadh, Saudi Arabia: RSP2023R27.

### Competing Interests

Authors declare that they have no competing interests.

### Author Contributions

- Md. Abdul Mannan conceived and designed the experiments, performed the experiments, analyzed the data, prepared figures and/or tables, authored or reviewed drafts of the article, and approved the final draft.
- Amir Yasmin conceived and designed the experiments, performed the experiments, analyzed the data, prepared figures and/or tables, authored or reviewed drafts of the article, and approved the final draft.
- Umakanta Sarker analyzed the data, authored or reviewed drafts of the article, and approved the final draft.
- Nasimul Bari performed the experiments, prepared figures and/or tables, and approved the final draft.
- Dipanjoli Baral Dola performed the experiments, prepared figures and/or tables, and approved the final draft.
- Hirokazu Higuchi analyzed the data, authored or reviewed drafts of the article, and approved the final draft.
- Sezai Ercisli analyzed the data, authored or reviewed drafts of the article, and approved the final draft.
- Daoud Ali analyzed the data, authored or reviewed drafts of the article, and approved the final draft.
- Saud Alarifi analyzed the data, authored or reviewed drafts of the article, and approved the final draft.

### Data Availability

The raw data are available in the Supplemental Files.

## Supplemental Information

Supplemental information for this article can be found online at http://dx.doi.org/10.7717/peerj.15588#supplemental-information.

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
