# Peer review of "Biostimulant red seaweed (Gracilaria tenuistipitata var. liui) extracts spray improves yield and drought tolerance in soybean"

_PeerJ, doi:10.7717/peerj.15588_

## Round 0.1 · original submission · Major Revisions

The study has some limitations, including a small sample size, lack of statistical analysis, and insufficient information on the extraction method, seaweed extract composition, generalizability, and environmental impact. The authors also need to address the comments of reviewers.

Reviewer 1 ·

Basic reporting

The manuscript “Biostimulant red seaweed (Gracilaria tenuistipitata var. liui) extracts improve yield and drought tolerance in soybean” deals with valuable insights into the potential of red seaweed extracts as biostimulants for enhancing soybean growth and yield under drought stress. The results show that seaweed extracts considerably increased soybean growth and yield under both well-watered and drought circumstances, with the greatest effect seen at a concentration of 10.0% v/v. The study also provides evidence that seaweed extracts can improve plant growth and physiology by increasing leaf water and chlorophyll content and reducing water saturation deficit. One potential limitation of the study is the use of only one species of seaweed. It would be useful to investigate the effects of seaweed extracts from other species on soybean growth and yield under similar conditions to assess the generalizability of the findings. The author needs to revise the manuscript before the final decision is made.

Comments
• Ln 32-35: Please rewrite the line.
• In the abstract section, provide future impact in one line.
• The introduction section seems to be written very haphazardly. This section needs to be written more clearly and more deeply to justify the study.
• LN 64-70: the author has provided many self-citations. The must refrain from using forced self-citation in the manuscript. Use only required and important citations.
• Similarly, check citation from line 72-82.
• LN 100: Use recent reference and use recent reference.
• The objective of the study is very poorly written. Kindly rewrite it.
• LN 132: Why heat treatment was needed.
• LN 173: Please describe the method in detail.
• To corroborate the conclusions drawn from the results, it would be helpful to offer some statistical analysis of the data, such as analysis of variance (ANOVA) or t-tests. This would strengthen the study's methodology and increase the validity of its findings.
• It's ridiculous that the author has provided 44 references in the manuscript. Its too much for a manuscript to have self-citation.
• LN 320-327: Please rewrite the sentence.
• The conclusion section need to be revisited and rewritten.
• Please improve the quality of the figures. It looks too blur.
• The caption of the figure can be more improved
• Kindly improve the English language in the introduction and discussion section.
• Explanation of significant values: In the tables, it would be helpful to provide a legend or footnote that explains the significance of the asterisks or other symbols used to indicate significant values. You can mention the statistical significance levels associated with each symbol, such as * for p < 0.05 or ** for p < 0.01.
• Future directions: In the conclusion section, you can add a few sentences about the potential directions for future research. This can include areas where the findings can be further explored or expanded upon, or potential applications of the results in real-world contexts. It can also be helpful to identify any limitations of the study and suggest ways to address them in future research.

Experimental design

Please see the comments above

Validity of the findings

Please see the comments above

Additional comments

Please see the comments above

Reviewer 2 ·

Basic reporting

This article is well written and have scientific implications.
There are some minor corrections needed to improve the manuscript.
1. Please avoid the references bellow 2015
2. Please discuss the finding with proper reasons, many places the justifications are not sufficient.
3. Please concise the materials and methods, it became so lengthy.
4. In all the table, please add what is the value after +- indicates in foot note.

Experimental design

1. It is advised to the authors to give the ANOVA table at p<0.01 instead off p<0.05
2. The result table is not seems as two factor CRD. It seems its normal CRD.

Validity of the findings

No nomments

Annotated reviews are not available for download in order to protect the identity of reviewers who chose to remain anonymous.

Reviewer 3 ·

Basic reporting

The literature should be more up-to-date and sentences need to be recasted.

Experimental design

Well defined

Validity of the findings

No comment

Additional comments

The research is valid and requires further confirmation.

Annotated reviews are not available for download in order to protect the identity of reviewers who chose to remain anonymous.

---

## Round 0.2 · accepted · Accept

The manuscript can be accepted in its current form.

Reviewer 1 ·

Basic reporting

The authors changes in the manuscript as per the suggestion received from reviewer.

Experimental design

The authors changes in the manuscript as per the suggestion received from reviewer.

Validity of the findings

The authors changes in the manuscript as per the suggestion received from reviewer.

Reviewer 2 ·

Basic reporting

The current form of the manuscript may kindly be accepted for publication.

Experimental design

Appropriate

Validity of the findings

Recognized and correctly represented.